# Pathway-Specific Insights into Colorectal Cancer Through Comprehensive Multi-Omics Data Integration

**DOI:** 10.3390/biology14050468

**Published:** 2025-04-25

**Authors:** Tayyip Karaman, Sinem Oktem Okullu, Günseli Bayram Akçapınar, Osman Ugur Sezerman

**Affiliations:** 1Department of Medical Biotechnology, Institute of Health Science, Acibadem Mehmet Ali Aydinlar University, Atasehir, Istanbul 34752, Turkey; tayyipkaramannn@gmail.com (T.K.); gunseli.akcapinar@acibadem.edu.tr (G.B.A.); 2Department of Medical Microbiology, School of Medicine, Acibadem Mehmet Ali Aydinlar University, Atasehir, Istanbul 34752, Turkey; sinem.oktem@acibadem.edu.tr; 3Department of Biostatistics and Medical Informatics, School of Medicine, Acibadem Mehmet Ali Aydinlar University, Atasehir, Istanbul 34752, Turkey

**Keywords:** multi-omics data integration, microbiome, transcriptome, metabolic pathways, colorectal cancer

## Abstract

Multiple parameters have a direct or indirect impact on cancer-like complex disease progression. Multi-omics data analysis has paved the way for understanding the complex interactions between these parameters, which play crucial roles in disease occurrence. Our study furthers the understanding of how microbiomes and the transcriptome interact with each other in colorectal cancer and provides pathway-specific approaches to determine how microbial and host mechanisms work in cancer.

## 1. Introduction

Developments in next-generation sequencing and bioinformatics have revolutionized our ability to understand complex biological systems and their relationships with disease occurrence [1,2,3]. Multi-omics data integration is a computational approach that combines multiple layers of omics datasets. It has emerged as a significant tool to unravel biological mechanisms, providing a comprehensive understanding of the molecular changes in disease occurrence [4].

Omics studies, such as metagenomics, genomics, transcriptomics, proteomics, metabolomics, etc., provide remarkable insights to comprehend disease states [5,6]. A single-omics approach is insufficient to comprehensively evaluate complex diseases, such as cancer [7,8], because complex diseases, like cancer, need the combination of the omics study outputs to provide an evaluation of the disease from all perspectives [9]. The most preferred multi-omics integration techniques are PCA (principal component analysis), CCA (canonical correlation analysis), CoIA (co-inertia analysis), MFA (multivariate factor analysis), PMA (penalized multivariate analysis), PLS (partial least squares), RFE (recursive feature elimination), graph-fused methods, LASSO (least absolute shrinkage and selection operator) penalization methods, Bayesian classification methods, and regression methods [10,11,12,13]. These techniques generally focus on collecting the most explanatory features from different layers of omics datasets and merging all this information to obtain interactions. Despite the dramatic increase in the number of integration studies in recent years, it has remained difficult to obtain a gold standard. This is because handling a large number of datasets containing a huge number of variables is challenging [14,15]. However, the results obtained from the developed multi-omics analysis packages provide promising results, especially in explaining complex cancers [16,17,18]. In research conducted by Bisht et. al., RFE, the Bayesian additive regression trees method, and Bayesian classification were used for the integration of omics datasets, and they discovered that the *Kiaa1199*, *Cdh3*, *Guca2b*, *Lgals4*, *Ca7*, *Nrc3c2*, *Abcg2*, and *Aqp8* genes were associated with cancer pathophysiology [19]. In addition, they also discovered that *Staphylococcus*, *Blautia*, and *Roseburia* interact with cholesteryl ester metabolites, which are prominent in colorectal cancer patients. All findings have shown remarkable colorectal cancer hallmarks, such as resistance to apoptosis, angiogenesis, and cell evasion, increasing our understanding of molecular-level effects on the progression of colorectal cancer.

Colorectal cancer (CRC) is the third most common cancer type worldwide and demonstrates dynamic variations due to environmental effects such as geography, climate, lifestyle habits, diet, etc. [20,21]. Recent research efforts have been focused on finding potential biomarkers for CRC based on stage information. In a study published in 2017, the transcriptome dataset of 377 patients (data obtained from the Cancer Genome Atlas) was analyzed, and the expression levels of the *Nek4*, *Rnf34*, *Hist3h2bb*, *Nudt6*, *Lrch4*, *Glb1l*, *Hist2h4a*, *Tmem79*, *Amıgo2*, *C20orf135*, and *Spsb3* genes were found to be significantly different between cancer stages (*p* = 0.0001) [6]. CRC patients are classified into four groups, known as consensus molecular subtypes (CMS1, CMS2, CMS3, and CMS4) [22,23]. Consensus molecular subtypes (CMSs) of colorectal cancer are stratified based on key biological characteristics, including immune infiltration and activation, signaling pathway dynamics, metabolic dysregulation, and stromal involvement. These subtypes provide critical insights into the heterogeneity of CRC, reflecting differences in tumor biology, patient prognosis, and therapeutic response. While transcriptomic profiling remains the predominant method for CMS classification, emerging research is exploring microbiome-based predictive models, leveraging host–microbiome interactions to infer CMS groups with potential implications for precision oncology. A study revealed the association of microbial changes between the CMS groups: *Fusobacteria* and *Bacteroidetes* were found to be enriched in the CMS1 subgroup, for example, whereas *Firmicutes* and *Proteobacteria* decreased. In addition, *Fusobacterium hwasookii* and *Porphyromonas gingivalis* demonstrated the highest enrichment in the CMS1 subgroup, while *Selenomonas* and *Prevotella* species exhibited a significant increase in the CMS2 subgroup [24]. This grouping approach is important for accurate diagnosis and selecting the best treatment methodology. Studies that focused on the assessment of the gut microbiome and the host transcriptome have provided in-depth knowledge about colorectal cancer occurrence. However, it also needs the integration of omics results to provide pathway-specific interactions.

In this study, we aimed to integrate the microbiome, transcriptome, and microbial pathway datasets obtained from patients with CRC to unveil the specific interactions between the host genome and microbial community. The results of the integration were consolidated by the host pathway dataset obtained from transcriptome data and databases such as bencard, MetaCyc, NCBI, KEGG (Kyoto Encyclopedia of Genes and Genomes), and UniProt (Universal Protein Knowledgebase) [25,26,27,28,29]. Our analysis consisted of two approaches, one focusing on the integration of all information from the omics analysis results to provide a more comprehensive perspective and not to neglect any potential interaction. The second one was more specific; it only focused on the integration of the statistically significant outputs of the omics analysis. Both approaches presented non-negligible outcomes by providing latent interactions between the datasets.

## 2. Materials and Methods

### 2.1. Patient Cohort and Experimental Design

Transcriptome and microbiome 16S rRNA raw datasets were obtained from a recent study titled ‘Distinct gut microbiome patterns associate with consensus molecular subtypes of colorectal cancer’ published by Rachel V. Purcell et al. in 2017 [24]. In that study, tumor samples were collected from 34 colorectal cancer patients (20 female, 14 male); the age ranged from 44 to 88 years (mean: 74 ± 8.4). In addition to age and gender, the metadata also contained ‘site’ data, indicating where the tumor sample had been taken from (1 rectal sample and 33 colonic samples), ‘side’ data, referring to the side of the body the tumor sample was taken from (21 from the right side and 12 from the left side), ‘stage’ data, referring to the stage of the cancer patient (5 in stage I, 14 in stage II, 13 in stage II, and 1 in stage IV), and ‘CMS’ data (6 in the CMS1 group, 13 in the CMS2 group, 9 in the CMS3 group, and 5 unclassified (UC) samples). Both the transcriptome and microbiome raw datasets were analyzed again, including pathway analysis. Microbial pathway abundance data were constructed based on the pathway analysis of the 16S rRNA. Microbial pathway abundance data, transcriptome data, and microbiome datasets were integrated using two approaches: (1) normalization and sparse partial least squares discriminant analysis and (2) normalization, feature selection, and the N-integration method indicated in Figure 1. The dataset obtained from the transcriptome analysis, bencard database, and MetaCyc databases were used for the validation of the information obtained from the integration of the three datasets.

### 2.2. Microbiome and Pathway Analysis

16S rRNA analysis was performed to determine the microbial communities in the tumor samples. Raw datasets that were generated by the MiSeq platform (Illumina, San Diego, CA, USA) contained 250 bp long reads that covered the V3–V4 region of the 16S rRNA gene. FLASH (ver. 1.2.11) was used for merging the forward and reverse reads. Preprocessing analysis was conducted using QIIME2 (ver. 2023.5) [30]. Quality assessment and removal of chimeric sequences were performed using DADA2 (ver. 3.19) [31]. The taxonomic classification was performed using the q2-feature-classifier plugin within the QIIME2 framework, leveraging the Greengenes database as a reference. Operational taxonomic unit (OTU) tables were subsequently generated. Differential abundance analysis was conducted using ANCOM-BC (ver. 3.19).

The biological observation matrix files derived from the QIIME2 pipeline were utilized for microbial functional pathway prediction analysis, which was carried out using the PICRUST2 (ver. 2.4.1.) plugin on the QIIME2 analysis platform [32]. The KEGG orthology database was used for the estimation of molecular functions. The 16S rRNA dataset and microbial pathway abundance dataset were saved for further integration analysis.

### 2.3. Transcriptome and Pathway Analysis

Tumor samples were sequenced using the, HiSeq platform (Illumina, San Diego, CA, USA) generating 125 bp paired-end reads. Adapter sequences were removed using Fastq-mcf (v. 1.1.2.537), followed by quality-based trimming of long reads, the removal of reads shorter than 50 bp, and the exclusion of low-quality reads using SolexaQA++ (v. 3.1.6). The filtered reads were aligned to the human genome reference GRCh38 using the STAR mapping tool (v. 2.5.2b). The resulting alignments were processed using Samtools (v. 1.3.1) for read merging and quantification via htseq-count.

Raw read counts were normalized to transcripts per million (TPM) using the DESeq2 package (v. 1.10.1) in R (version 4.3.1; R Foundation for Statistical Computing, Vienna, Austria). RNA-seq analysis culminated in pathway enrichment analysis using the ‘pathfindR’ R package (v. 2.1.0) in R [33]. Pathway annotations were retrieved from the KEGG database, with significant pathways filtered based on a false discovery rate (FDR) threshold of 0.05. Pathways meeting the FDR-adjusted p-value criteria were selected for the downstream analyses. Transcriptome datasets were reserved for further integration analysis, whereas host (human) pathway datasets were kept for the validation of the results obtained from the multi-omics data integration.

### 2.4. Data Standardization and Preprocessing

The 16S rRNA microbiome data (at each taxonomic level), microbial pathway abundance data, and transcriptome data were normalized using the centered log ratio (CLR) normalization method. Sparse partial least squares discriminant analysis (sPLS DA) was applied to the data to fulfill our first approach in the multi-omics integration. Preprocessing was applied to each dataset to improve the accuracy of the machine learning model, and the correlation coefficient was obtained at the end of the first approach integration. Samples that belonged to the UC group were excluded from the dataset directly using the dplyr (ver. 1.1.4.) and tidyverse packages (ver. 2.0.0.). Feature selection was then applied based on the significance of the variables in the datasets. First, the Shapiro–Wilk test was applied to check the normality of the data, and then the Kruskal–Wallis test was performed. Variables that were found to be nonsignificant were filtered out from the datasets. We constructed new microbiome (phylum level *n* = 5, where n refers to the number of variables), transcriptome (*n* = 851), and microbial pathway abundance (*n* = 26) datasets containing only significant variables at the end of the preprocessing step. CLR normalization was applied to the new datasets to be used in the N-integration method.

### 2.5. Data Integration Using sPLS Analysis

Sparse partial least squares discriminant analysis (sPLS-DA) was performed using the mimics package (version 6.10) in R for the 16S rRNA microbiome, the transcriptome, and the microbial pathway abundance datasets as pairs to understand the association between them [34]. PLS models were tuned based on 10-fold cross-validation, and the parameters were used as defaults applied for two matrices, X and Y, which corresponded with the omics datasets. Selected explanatory variables were used for the visualization of the CMS groups via PCA, and the correlations between the variables that were selected from different datasets were displayed as heatmaps constructed based on hierarchical clustering with the Euclidean distance method.

### 2.6. Data Integration Using N-Integration

The data integration analysis for the biomarker discovery (DIABLO) method, also known as the N-integration method, was used for the specific detection of biomarkers. Preprocessed datasets were used to increase the accuracy of the model compared to the previous approach. The supervised model was tuned ten times based on 10-fold cross-validation. Most explanatory variables obtained from the constructed components were separated and underwent correlation. Heatmaps based on hierarchical clustering with the Euclidean distance method and network visualization based on correlations between variables were used to illustrate the associations between different omics datasets.

### 2.7. Statistical Analysis

All statistical analyses were conducted in R (version 4.4.1; R Foundation for Statistical Computing, Vienna, Austria) [35]. Data distribution was assessed using the Shapiro–Wilk test. For subgroup comparisons, the Kruskal–Wallis test and pairwise Wilcoxon tests were applied, with multiple testing corrections performed using the Benjamini–Hochberg method. An FDR-adjusted p-value threshold of <0.05 was set for statistical significance. Machine learning methodologies were implemented utilizing the mixOmics (v. 6.10) and caret (v. 6.0.94) packages in R, facilitating advanced data integration and predictive modeling workflows.

## 3. Results

### 3.1. Microbiome Analysis and Functional Pathway Prediction

The results of the 16S rRNA microbiome analysis indicated that CRC patients that belonged to the CMS1 group had enriched levels of *Fusobacteria* (*p* < 0.05) and *Bacteroidetes* (*p* > 0.05) and decreased levels of *Firmicutes* (*p* > 0.05) and *Proteobacteria* (*p* < 0.05) at the phylum level (*p* < 0.05) (Figure 2). Although the genus-level results were not visualized, our analysis revealed a significant increase in the relative abundance of *Prevotella* in the CMS1 group, whereas *Bacteroides* was markedly enriched in the CMS2 and CMS3 groups. The most abundant genus was *Bacteroides* in each CMS group (48% in CMS1, 65% in CMS2, and 29% in CMS3), followed by *Faecalibacterium* as the second most abundant genera in CMS1 and CMS2 (15% in CMS1 and 4% in CMS2). *Clostridium* and *Faecalibacterium* had almost the same relative abundance in the CMS3 patient group (both 5.5%). Data normality was checked by the Shapiro test, and the nonlinear distribution of variables was confirmed. Therefore, the pairwise Wilcoxon test was applied to each CMS group to observe the significant differences between each group. The results revealed no significant differences between the CMS subgroups. The statistical results obtained from the pairwise Wilcoxon test are provided in Table 1. The Picrust2 plugin was used for the functional characterization of the OTU tables. Pathway information was obtained from the KEGG database. A total of 478 pathways (shown in Appendix A) were found by pathway prediction analysis, and univariate feature selection was performed to determine the significant variables (pathways) that decreased to 26 after selection. These pathways were kept for the multi-omics integration analysis.

### 3.2. Transcriptome and Pathway Enrichment Analysis

A volcano plot was constructed using the results generated from RNA-seq analysis. The *Rn7sk*, *Rpph1*, *Rn7sl1*, *Rn7sl2*, *Znf460*, *Snord17*, *Snora73b*, *H1-4*, *H4c12*, and *H3c7* genes were discovered to be highly upregulated in the tumor tissue collected from colorectal cancer patients (Figure 3A). PCA was also carried out for the RNA-seq results. Variables included in the transcriptome data did not cluster each CMS group properly (Figure 3B). RNA-seq analysis was followed by pathway enrichment analysis to observe which genes were upregulated or downregulated in their related pathways. A total of 179 pathways (from the KEGG database) were found to be associated with over 5000 genes that are either downregulated or upregulated, as indicated in Appendix A. The pathway enrichment results were not used for the integration of the omics datasets; they were used for the validation of the selected genes attained from sPLS and the DIABLO multi-omic integration analysis.

### 3.3. Integration of Microbiome and Transcriptome Datasets Using the sPLS Method

Pairwise multi-omics integration was carried out using the sPLS method for the dataset generated as a result of 16Sr RNA and transcriptome analysis. This first approach of our integration analysis aimed to find an association by including all variables to allow us to observe the relationships comprehensively. Variables that were selected by regression were used to create a heatmap to illustrate the correlations between each dataset.

The heatmap analysis revealed intricate correlations between phylum-level microorganisms and gene expression profiles derived from transcriptomic data (Figure 4A). Excluding *Spirochetes* and *Verrucomicrobia*, the heatmap directly discriminated patients into two groups. The first cluster included the *Klf3*, *Cldn7*, *Plekhg3*, *Mall*, *Rna5-8n2*, *Snord10*, *Sytl2*, *Dnm2*, *Cxadr*, *Tff3*, *Serpina1*, *Fam120a*, *Rtn4*, *Mıb1*, *Oip5-as1*, *S100a6*, and *H1-5* genes, along with the phyla *Actinobacteria*, *Cyanobacteria*, *Tenericutes*, *TM7*, *Bacteroidetes*, *Firmicutes*, and *Proteobacteria*. The second group encompassed the genes *Adar*, *Csnk2a1*, *Rpl37*, *Ahctf1*, *Odc1*, *Nup98*, *Ccdc14*, *Znf292*, *Ccnd2*, *Igkc*, *Prdx5*, *Gpsm2*, *Gpx2*, *Pan3*, *Tkt*, *Rpl22*, *Ahcy*, *Nme2*, *Rbm12*, *Cct6a*, *Rplp2*, and *Slc7a1*, together with the bacterial phyla *Synergistetes* and *Fusobacteria*. Notably, a strong positive correlation was observed between the phylum *Spirochetes* and the expression of the *Oip5-as1*, *S100a6*, *H1-5*, and *Adar* genes (r = 0.71), indicating a potential regulatory relationship. Likewise, *Fusobacteria* exhibited a significant positive association with the *Klf3*, *Cldn7*, *Sytl2*, *Dnm2*, and *Tff3* genes (r = 0.70), underscoring its potential involvement in related molecular pathways. Correlations with coefficients below 0.5 were excluded from further investigation. Negative correlations were also identified. *Spirochetes* showed a strong negative relationship with *Tff3*, *Serpina1*, and *Prdx5* (r = −0.71). In addition, *Fusobacteria* was inversely correlated with the *Odc1*, *Nup98*, *Ccdc14*, *Ahcy*, and *Cct6a* genes (r = −0.69). Furthermore, *Synergistetes* showed negative associations with *Rpl22*, *Ahcy*, and *Gpsm2*.

These findings highlight the complex interplay between microbial taxa and host gene expression, emphasizing key correlations that may provide insights into functional and pathological mechanisms. The PCA that was conducted on selected variables obtained from the microbiome and transcriptome datasets did not manage to discriminate between the CMS groups (Figure 4B).

### 3.4. DIABLO Integration for the Preprocessed Microbiome, Transcriptome, and Microbial Pathway Datasets

Unclassified samples were extracted from the microbiome, transcriptome, and microbial pathway abundance datasets, and nonsignificant variables from each dataset were excluded via the pairwise Wilcoxon test based on the CMS subgroups. This second approach with cleanup preprocessing was applied to increase the model accuracy and to obtain more precise correlations by minimizing data heterogeneity. DIABLO was carried out to integrate the omics data. Preprocessed data underwent sPLS-DA first, and the components were constructed with variables from each omics block. The majority of the explanatory variables were identified using a supervised learning model, and these variables were consolidated into components. For the downstream analysis, the most informative components (the first and second components) were selected. To evaluate the discriminative power of the variables comprising these components, a sample plot was employed, revealing their ability to delineate and characterize group differences effectively. Notably, the variables derived from the transcriptome block demonstrated a robust capacity to capture the segregation and intricate relationships among the molecular subtypes (Figure 5).

Significant features were selected as explanatory variables for the classification of CMS subgroups as a result of the sPLS-DA regression model. *Rothia* from the microbiome data and the L-lysine and CRNFORCAT pathways were selected from the microbial pathway data to explain the patient status of the CMS1 subgroup. The model did not identify any transcriptomic features as discriminative for the CMS1 subgroup. However, for the CMS2 subgroup, the transcriptomic analysis highlighted key genes, including *Top2a*, *Hsp90ab1*, *Ncl*, *Ahcy*, *Hspd1*, *Psma7*, *Eif2s2*, *Cbx3*, *Cse1l*, *Lbr*, and *Tpx2*, as critical for distinguishing this molecular subtype. In addition, the microbial pathway analysis identified the p-cymene degradation pathway and the glutaryl CoA degradation pathway, while *Dialister* from the microbiome data showed the highest explanatory value for classifying the CMS2 subgroup.

For the CMS3 subgroup, the model pinpointed *Atp8a1*, *Akap13*, and *Reg4* from the transcriptomic dataset as highly informative. Furthermore, microbial taxa such as *Desulfovibrionaceae*, *Gamellaceae*, and *Paraprevotella* were identified as significant contributors to the microbiome dataset. From the microbial pathway dataset, the myo-inositol degradation pathway was selected for its relevance in explaining the CMS3 subgroup. These multi-omics features collectively provide a comprehensive framework for understanding the molecular and microbial underpinnings of the CMS subgroups (Figure 6).

A circle plot was used to observe the relationships between the variables that were obtained from different omics datasets (Figure 7). The variable coordinates were determined by the correlation of components 1 and 2 obtained from features extracted by 10-fold cross-validation. The features that constructed components 1 and 2 were plotted in the coordinate system. The proximity of the variables at the poles indicated a strong biological relationship.

Further investigation was made to understand the relationships between the variables. The associations were illustrated by network visualization and a Circos plot. The glutaryl CoA degradation and p-cymene degradation pathways were positively associated with *Ahcy*, *Eif2s2*, *Hsp90ab1*, *Psma7*, *Lbr*, *Rpl7l1*, *Cse1l*, *Cbx3*, *Ncl*, *Hspd1*, *Tpx2*, and *Top2a* genes. In particular, the association of these pathways (glutaryl CoA degradation and p-cymene degradation pathways) with the *Cse1l* and *Tpx2* genes was very high (r = +0.94) (Figure 8).

A Circos plot was generated to examine the correlations between the variables attained from each omics dataset more closely (Figure 9). The plot indicated the positively and negatively correlated features by connecting them with lines. In our study, the myo-inositol degradation pathway exhibited a significant negative association with the genes *Top2a*, *Tpx2*, *Hspd1*, and *Cse1l*. Similarly, the glutaryl CoA and p-cymene degradation pathways were negatively correlated with *Paraprevotella* (from the microbiome dataset) and the transcriptomic genes *Akap13*, *Reg4*, and *Atp8a1*.

Conversely, the methionine acetate pathway demonstrated a positive correlation with the *Sema5a* gene. Positive associations were also identified between the glutaryl CoA degradation pathway, the p-cymene degradation pathway, and the *Dialister* genus. In addition, *Ruminococcaceae* and *Lachnospiraceae* were positively linked to *Sema5a*, whereas *Dialister* showed a positive correlation with *Cse1l*. In contrast, *Paraprevotella* displayed a negative association with the genes *Tpx2* and *Cse1l*, underscoring a complex interplay between these microbial and transcriptomic elements within the dataset. These findings provide nuanced insights into the interconnected pathways and their relationships across multiple data modalities.

## 4. Discussion

A huge number of variables, contents, and structures of different omics datasets have been an issue to deal with. Nevertheless, the number of multi-omics integration studies increased because they provided great insights into understanding complex mechanisms. In our study, we aimed to produce interactions of transcriptome, microbiome, and microbial pathway datasets obtained from colorectal cancer patients. Pathway enrichment analysis was performed for transcriptome data, and human pathways associated with upregulated and downregulated genes were recorded. Both microbial pathway datasets obtained from 16SrRNA analysis and the pathway enrichment table results obtained from transcriptome analysis gave promising outputs in mechanisms of colorectal cancer progression. The Metacyc, Biocyc, Genecard, and KEGG databases were used to evaluate the functional interactions that came from the multi-omics integration analysis.

### 4.1. sPLS for Multi-Omics Data Integration

Partial least squares (PLS) was selected among numerous available approaches due to its proven utility in multi-omics integration studies and its robustness for continuous development. The sparse partial least squares (sPLS) provided a comprehensive overview of the data, despite the known limitation of reduced explanatory power for individual components. This limitation arises from the method’s emphasis on retaining as many variables as possible to maximize covariance between datasets. Nevertheless, the results derived from the subsequent correlation analysis following sPLS regression were highly interpretable and biologically relevant.

Notably, the analysis revealed a remarkable association between the *Cldn7* gene and *Fusobacteria*. *Cldn7*, a critical member of the claudin family, plays a fundamental role in the structural integrity of tight junctions. The reduced expression level of this gene has been strongly linked to the progression of cancer [36]. *Fusobacteria* species, frequently detected in colorectal cancer, are known to disrupt the intestinal barrier. Previous studies have demonstrated that *Fusobacterium nucleatum* compromises the oral mucosal epithelial barrier by inhibiting *Cldn4*, another claudin family member [37,38]. Our findings suggest that the observed strong correlation between *Fusobacteria* and *Cldn7* may indicate a similar mechanism, leading to tight junction disruption and subsequent barrier collapse.

In addition, *Klf3* was identified as positively correlated with *Fusobacteria*. *Klf3* is a transcription factor that plays a pivotal role in activating WNT1 and WNT/β-catenin signaling pathways, which are key drivers of colorectal cancer progression [39]. Supporting this observation, previous research has demonstrated that *Fusobacterium nucleatum* promotes colorectal cancer progression through Cdk5-mediated activation of the WNT/β-catenin pathway [40].

RNA-Seq enrichment analysis further supported these associations, revealing the downregulation of *Wnt2*, *Wnt3*, *Notum*, *Lrp5*, *Lrp6*, *Csnk2a2*, *Gsk3b*, *Chd8*, *Crebbp*, *Ep300*, *Tbl1xr1*, *Invs*, and *Plcb3* and the upregulation of *Wnt2b*, *Wnt5b*, *Serpinf1*, *Dkk1*, *Sfrp1*, *Sfrp2*, *Sfrp4*, *Rspo2*, *Rspo3*, *Fzd1*, *Fzd4*, *Frat1*, *Tcf7l1*, *Cby1*, *Sox17*, *Ctnnd2*, *Ccnd3*, *Prkaca*, *Skp1*, *Rbx1*, *Ror1*, *Prickle1*, *Daam2*, *Rhoa*, *Rac2*, *Mapk10*, *Ppp3ca*, and *Ppp3cb*, all of which are involved in the WNT signaling pathway.

Without the integrative capability of sPLS, the identification of the interplay between *Klf3* and *Fusobacteria*, or *Cldn7* and *Fusobacteria*, would have been overlooked. Specifically, the association of *Fusobacteria* with *Cldn7* suggests a potential role in tight junction destabilization, while its correlation with *Klf3* points to its involvement in WNT/β-catenin pathway dysregulation in colorectal cancer. These findings underscore the value of sPLS in revealing complex biological relationships, despite its lower explanatory power for individual components. By offering a comprehensive integrative perspective, sPLS provides essential insights for pathway-driven diagnostics, which are crucial for understanding multifaceted diseases such as cancer.

### 4.2. Impact of the Preprocessing Datasets

The first approach aimed to include all the variables so as not to miss any detailed information. Despite the fact that even potential outcomes were received by sPLS analysis, we still needed to get rid of the noise of the datasets to produce more significant interactions. Unclassified samples were extracted from the datasets, and the Wilcoxon test was performed to keep only significant features. This preprocessing provided a reduction in noisiness in data, and performing DIABLO for this preprocessed data demonstrated the increased accuracy of the model. It also increased the correlation coefficient values obtained from the correlation of the variables constructed in the components of the model. In particular, explanatory variables obtained from the transcriptome dataset provided the best discrimination of CMS subgroups.

### 4.3. Multi-Block sPLS for Multi-Omics Data Integration for Preprocessed Datasets

As we interpret the results from the microbial pathway point, we discovered that the glutaryl CoA degradation and p-cymene degradation pathways were positively associated with *Ahcy*, *Eif2s2*, *Hsp90ab1*, *Psma7*, *Lbr*, *Rpl7l1*, *Cse1l*, *Cbx3*, *Ncl*, *Hspd1*, *Tpx2,* and *Top2a* genes (r > 0.65). Particularly, the correlation of these pathways with *Cse1l* and *Tpx2* genes was significantly high (r = 0.94). The p-cymene degradation pathway demonstrates an anti-tumor influence on cancer patients [41,42]. The mechanism behind this is thought of as a reduction in inflammatory factors, like IL-1 and LEP, whereas the upregulation of IL-6 promotes the growth of *Bifidobacteria* and *Clostridium IV* [43]. Another prominently identified microbial pathway was the glutaryl CoA degradation pathway, which plays a pivotal role in the metabolism of lysine, tryptophan, and hydroxylysine and has been recognized for its anti-tumor properties [44,45]. Prior research demonstrated that inhibition of this pathway induces NRF2 glutarylation, thereby enhancing its stability. This process subsequently activates the ATF4-ATF3 signaling cascade, ultimately leading to cancer cell apoptosis [46]. In the present study, the observed increase in p-cymene levels within tumor tissue suggests that this compound may represent a host immune response mechanism, potentially mediated by commensal microbiota to counteract tumor development and progression. Evidence of p-cymene-mediated inhibition of MMP-9 (matrix metalloproteinases) expression shows its anti-tumor activity. In addition to that, MMP-2 and MMP-9 inhibition plays an important role in the degradation of the extracellular matrix. Moreover, ERK1/2 and p38 MAPK signal pathways were also inhibited by p-cymene [41]. A direct or indirect impact of p-cymene on the inhibition of these genes emerges, promising therapeutic strategies. In our study, we have found that MMP-2 genes were downregulated in diabetic cardiopathy I, proteoglycans in cancer, estrogen signaling, fluid shear stress and atherosclerosis, relaxing signaling, bladder cancer, and GnRH signaling pathways (Appendix A). So, our results are strengthened by the literature and strongly propose the potential interaction of p-cymene on MMP-2 inhibition.

The *Hspd1* gene is a heat shock protein that is related to transcription and peroxisomal lipid metabolism [47]. The *Hspd1* gene has been identified as essential for maintaining metabolic fitness, with its activity being critical for sustaining cellular energy homeostasis [48]. A study published in 2021 demonstrated that loss-of-function mutations in *Hspd1* lead to a profound energetic collapse, which directly impairs the ability of cancer cells to proliferate and expand, thereby disrupting tumor progression [49]. *Cbx3* is taking a role in transcription, RNA polymerase, and promoter opening pathways. Also, studies indicate that this gene is overexpressed in a variety of cancers but is still a poor prognosis biomarker in cancer patients [50,51]. The *Cse1l* gene is a chromosome segregation 1-like gene that plays a role in the p53 effector and Golgi to endoplasmic reticulum traffic pathways. Researchers discovered that *Cse1l* directly promotes the development of tumors in various malignancies [52]. The *Lbr* gene is a Lamin B receptor that takes a role in the super pathway of cholesterol biosynthesis and transcription pathways. Its role in cancer was revealed as *Lbr* protects the genome from tumorigenesis and chromosomal instability [53]. The *Tpx2* gene is a microtubule nucleation factor found in microtubule organization pathways. It depicts upregulated expression level profiles in CRC patients [54]. In our study, significantly discovered microbial pathways had a notable protective role against cancer, whereas significantly found genes were generally oncogenes. This indicates the impact of the microbial community on tumor tissue to suppress tumor growth. Further research is needed to unravel the exact mechanism of action in these microbial pathways against cancer and their relation through these oncogenes.

A Circos plot was generated to observe the correlation between variables in a closer way. It indicates positively and negatively correlated features by connecting them with the lines with the correlation cutoff = 0.7. In our study, the myo-inositol degradation pathway was found to be negatively associated with *Top2a*, *Tpx2*, *Hspd1,* and *Csel1* genes. In addition to that, the glutaryl CoA degradation and p-cymene degradation microbial pathways demonstrated a negative association with *Paraprevotella* in the microbiome dataset and with the *Akap13*, *Reg4*, and *Atp8a1* genes in the transcriptome dataset. Conversely, the meth acetate pathway exhibited a positive correlation with the *Sema5a* gene. Additionally, the glutaryl CoA degradation and p-cymene degradation pathways were positively linked to the *Dialister* genus.

Notably, both *Ruminococcaceae* and *Lachnospiraceae* showed a positive correlation with the *Sema5a* gene, while *Dialister* was positively associated with the *Csel1* gene. An increased abundance of the genus *Dialister* has been reported in colorectal cancer patients [55]. Although the precise mechanisms underlying this association remain unclear, our findings indicate that the correlation between *Dialister* and elevated expression levels of *Csel1* may play a contributory role in the progression of colorectal cancer. In contrast, *Paraprevotella* exhibited a negative association with *Tpx2* and *Csel1* genes, suggesting potential functional interactions between microbial communities and host gene expression.

Components of the last integration model with the implementation of preprocessing datasets were constructed by the *Hspd1*, *Cbx3*, *Csel1*, *Lbr,* and *Tpx2* genes from transcriptome data, *Actinobacteria*, *Spirochaetes*, *Fusobacteria*, *Bacteroidetes,* and *Proteobacteria* from microbiome data, and the cholorthocleavage pathway, crnforcat pathway, myo-inositol degradation pathway, p-cymene degradation pathway, and glutaryl CoA degradation pathway from the microbial pathway abundance dataset. The contribution of the datasets to explain CMS groups was increased, and weights for the transcriptome, microbiome, and pathway reached 0.80, 0.73, and 0.60, respectively. The model that was designed to predict the CMS groups achieved over 95%. Correlation analysis made for the explanatory components and *Hspd1*, *Cbx3*, *Csel1*, *Lbr,* and *Tpx2* genes were discovered to be highly associated with p-cymene degradation and glutaryl CoA degradation pathways.

*Hspd1*, *Cbx3*, *Csel1*, *Ncl*, *Top2a,* and *Tpx2* genes and *Proteobacteria* indicated negative association. Moreover, *Proteobacteria* were also found to be negatively correlated with the p-cymene degradation and glutaryl CoA degradation pathways. P-cymene promotes the growth of the genus *Bifidobacterium* and the genus *Clostridium,* which belong to *Actinomycetota* and *Firmicutes,* respectively [43]. Including this negative association, all output shows reasonably significant outcomes, which are potential candidates taking part in colorectal cancer progression. However, experimental validation is still necessary to comprehend the exact mechanism of action at the molecular level.

## 5. Conclusions

Cancer is one of the most studied diseases worldwide; however, there is still no exact treatment approach to get over it. Rather than discovering new biomarkers, it is quite significant to understand a person’s specific outputs to unravel the mystery behind complex biological mechanisms in cancer, like complex diseases. Strong interactions of *Klf3* and *Cldn* genes with *Fusobacteria* were quite promising outputs to protect the tight junction structure on the intestinal barrier. Also, the role of p-cymene degradation and glutaryl CoA degradation pathways was discovered as very important defensive mechanisms against tumors. All these findings are quite promising to be validated in further experimental research. An increasing number of multi-omics integration approaches in research will evaluate the details of the molecular mechanisms and, eventually, we will be able to serve person-specific, pathway-dependent solutions to cure cancer.

## Figures and Tables

**Figure 1 biology-14-00468-f001:**
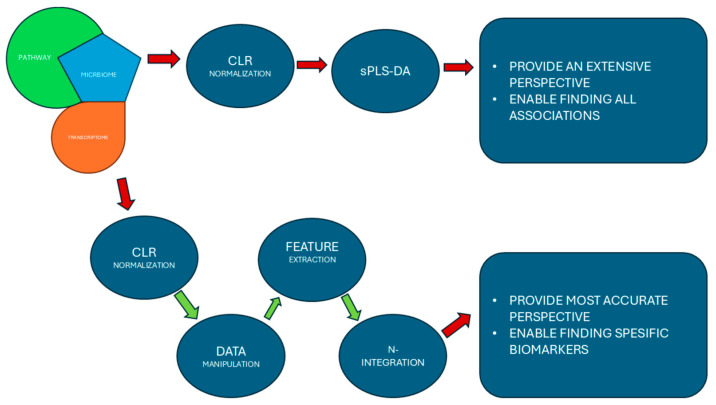
The chart demonstrates the experimental flow and advantages of both approaches at the end.

**Figure 2 biology-14-00468-f002:**
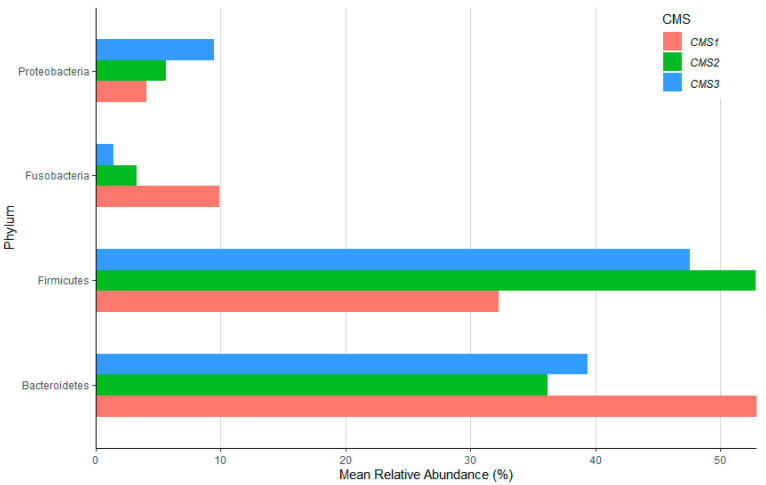
The boxplot shows the mean relative abundance percentile of *Proteobacteria*, *Fusobacteria*, *Firmicutes,* and *Bacteroidetes* between the CMS1 (red), CMS2 (green), and CMS3 (blue) groups.

**Figure 3 biology-14-00468-f003:**
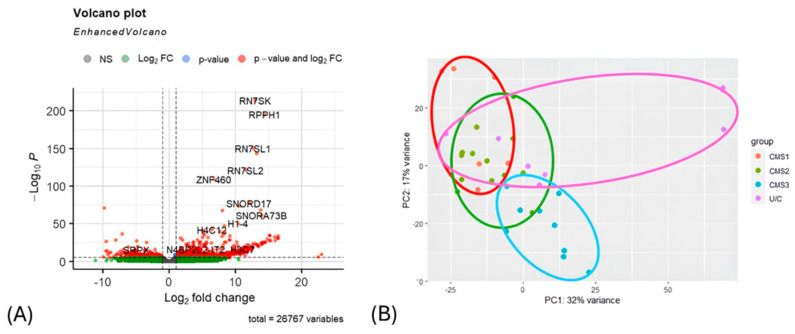
(**A**) Differentially expressed genes between CMS subgroups are illustrated in the volcano plot (*p* < 0.05). (**B**) The principal component analysis (PCA) visualization graph demonstrates the clustering of CMS subgroups based on RNA-seq data. Distinct groups are identified as follows: CMS1 is represented in red, CMS2 is denoted in green, CMS3 is illustrated in blue, and the UC subgroup is highlighted in purple.

**Figure 4 biology-14-00468-f004:**
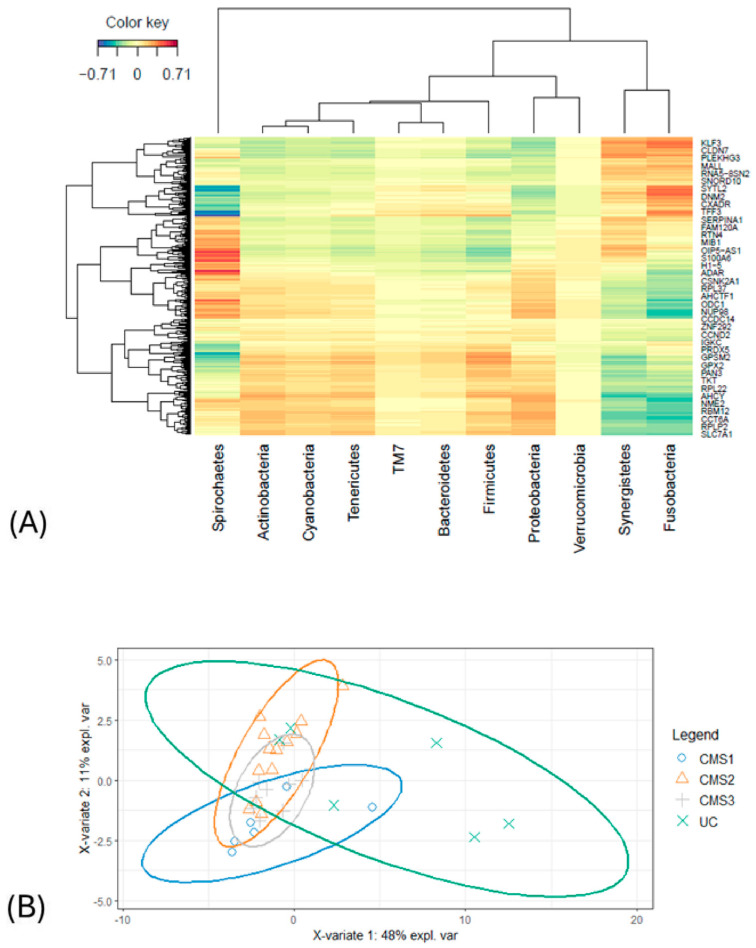
(**A**) The clustered heatmap from the sPLS-DA indicated a correlation between the microbiome (phylum) and transcriptome datasets (r = ±0.71). (**B**) The illustration projects how selected samples are clustered by PCA. Block X represents the microbiome dataset and block Y represents the transcriptome dataset. CMS1 is indicated in the blue circle, CMS2 is displayed in orange, CMS3 is shown in gray, and the UC group is represented in green.

**Figure 5 biology-14-00468-f005:**
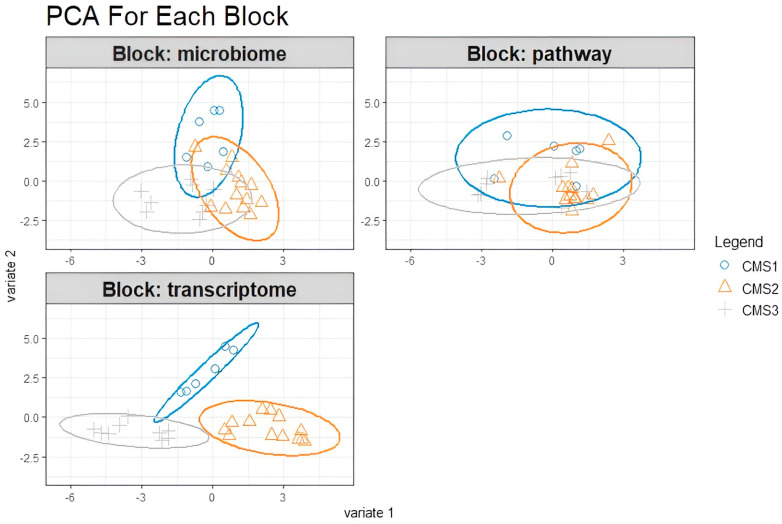
The samples were plotted according to the scores obtained from the first two components for each dataset. Variables are colored by CMS subgroup and classified into three classes: CMS1 (blue), CMS2 (orange), and CMS3 (gray).

**Figure 6 biology-14-00468-f006:**
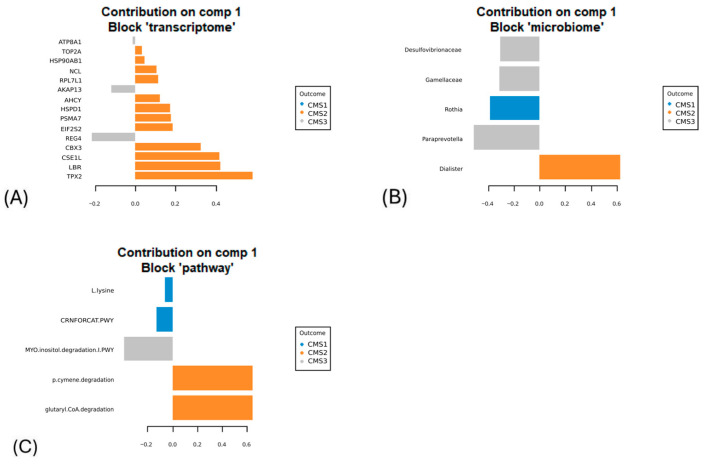
The variables are selected by the N-integration model, and the most explanatory contributors that constructed component 1 obtained from preprocessed (**A**) transcriptome, (**B**) microbiome, and (**C**) microbial pathway datasets are indicated in the figure (blue represents CMS1, orange represents CMS2, and gray represents CMS3).

**Figure 7 biology-14-00468-f007:**
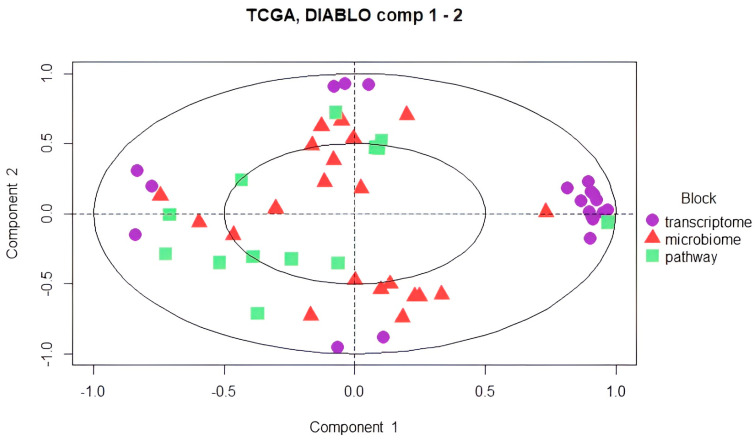
Circle plot showing the most predictive variables located at the poles. Purple dots represent features obtained from the transcriptome data, red triangles represent microbiome features, and green squares represent microbial pathways.

**Figure 8 biology-14-00468-f008:**
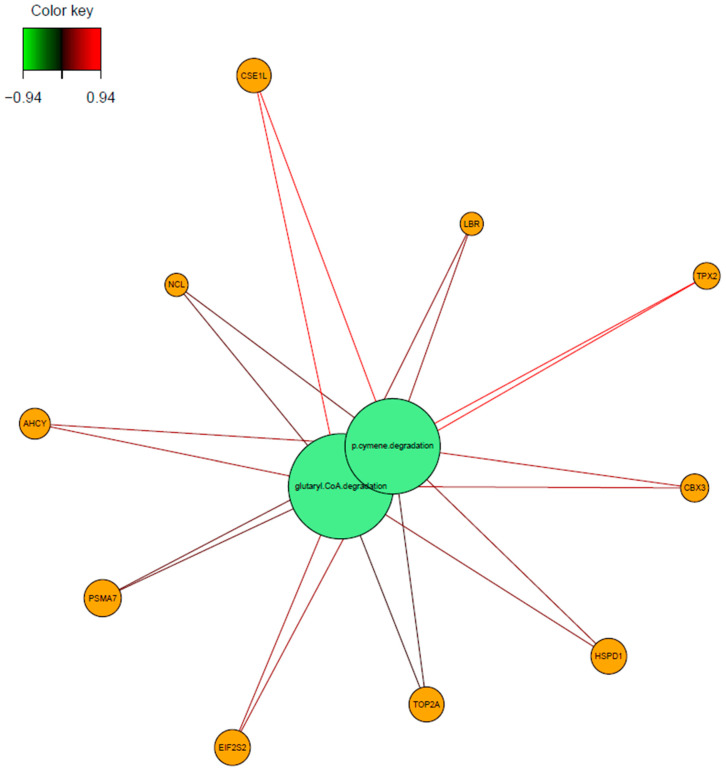
Relevance network for features selected by sPLS-DA performed on transcriptome and microbial pathway abundance data. Each node contains variables: green nodes represent the microbial pathways and orange nodes indicate the genes. The line that connects the features shows the relationship between the variables. Red lines indicate positive correlations and green lines indicate negative correlations.

**Figure 9 biology-14-00468-f009:**
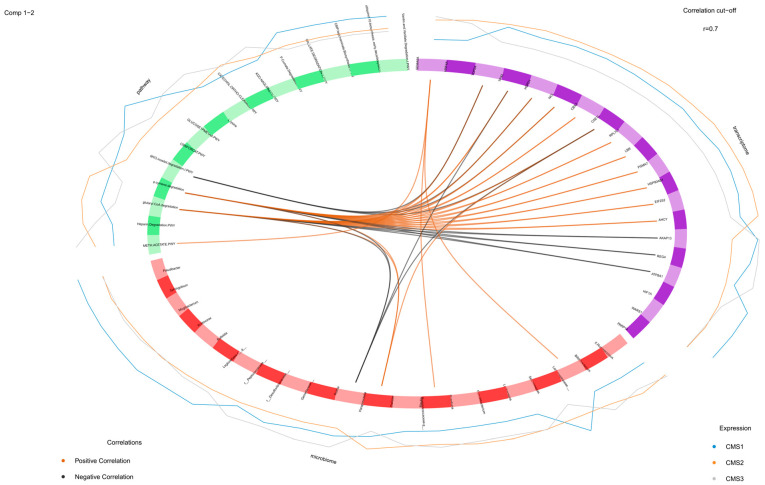
Correlations greater than 0.7 between variables obtained from the microbiome transcriptome and microbial pathway abundance are represented on the quadrants. The outer lines indicate the expression level of each sample group (CMS1, CMS2, and CMS3). The inner lines show the positive (orange) and negative (black) correlations among the variables.

**Table 1 biology-14-00468-t001:** The pairwise Wilcoxon results are indicated in the table.

Group 1	Group 2	H	*p*-Value	q-Value
*CMS1*	CMS2	0.769231	0.380455	0.570683
	CMS3	0.222222	0.637352	0.764822
	UC	3.692308	0.054664	0.231300
*CMS2*	CMS3	0.054627	0.8152	0.8152
	UC	1.3	0.254213	0.508426
*CMS3*	UC	3.125	0.0771	0.231300

## Data Availability

Transcriptome and microbiome 16S rRNA raw datasets were obtained from a recent study, “Distinct gut microbiome patterns associate with consensus molecular subtypes of colorectal cancer”, published by Rachel V. Purcell et al. in 2017, with permission from the authors. The datasets can be found in the PubMed database: https://pubmed.ncbi.nlm.nih.gov/31278253/ (accessed on 13 August 2021).

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
