# Peer review of "Pathway-Specific Insights into Colorectal Cancer Through Comprehensive Multi-Omics Data Integration"

_biology, 2025, doi:10.3390/biology14050468_

Round 1
Reviewer 1 Report
Comments and Suggestions for Authors
The paper presents a computational framework for integrating multi-omics data in colorectal cancer and highlights important pathway-specific interactions that could inform future diagnostics and treatment strategies. Since multi-omics studies in CRC are not entirely new, the focus on integrating microbial pathway data alongside host transcriptomics to indicate pathway-specific interactions should provide an innovative angle. However, the originality may be moderated by the fact that other studies have also pursued similar integrative approaches, thus, the novelty of current study may somehow suffer. There is a heavy reliance on correlation-based analyses, some references might suffer from outdated experimental studies. This manuscript is well-structured and methodologically detailed, offering a comprehensive view of the multi-omics integration process. It should be considered to meet conventional academic standards in terms of literature review, methodology, and statistical rigor. Further recommendation to make it a more suitable publication is that some of the figures should be replaced with the ones with higher resolution and adjust the inserted part below figure 2 caption with appropriate description.
Author Response
|
Comments 1: [Further recommendation to make it a more suitable publication is that some of the figures should be replaced with the ones with higher resolution and adjust the inserted part below figure 2 caption with appropriate description.]
|
|
Response 1: Thank you for pointing out your valuable comments. We agree with this comment. Therefore, we have upscaled the quality of the figures and specified the table.1 with an appropriate explanation. Hence, results become clearer to be understood. [Figure 2. Boxplot shows the mean relative abundance percentile of Proteobacteria, Fusobacteria, Firmicutes and Bacteroidetes between CMS1(red), CMS2(green) and CMS3(blue) groups.]
|

Reviewer 2 Report
Comments and Suggestions for Authors
The manuscript describes a multi-omics study using microbiome, transcriptome and microbial pathways datasets to understand correlations between gut microbial communities and the host mechanisms in colorectal cancer. Through the multi-omics study, the authors suggested Cldn7 gene that takes a role in intestinal barrier was associated with Fusobacteria. Klf3 was found to associate with upregulation of WNT1 and WNT/ β-catein pathways and also Fusobacteria. Another finding was that Glutaryl-CoA degra- 31 dation and p-cymene degradation pathways were associated with Ahcy, Eis2s2, Hsp90ab1, Psma7, Lbr, Rpl7l1, Cse1l, Cbx3, Ncl, Hspd1, 33 Tpx2, and Top2a genes.
The manuscript is well written, and the data provide insights into understanding of colorectal cancer and potential diagnostic potential.
Please find below with some minor comments:
- The table between line 247 and line 248 does not have a title or a table description.
- Figure 6 missing labeling such as A, B and C on the three graphs.
- Line 348 please undo the italicizing the word “and”.
Author Response
|
Comments 1: [The table between line 247 and line 248 does not have a title or a table description]
Response 1: Thank you for pointing out your valuable comments on our paper. We agree with this comment. Therefore, we have specified the table.1 with an appropriate explanation.
Comments 2: [Figure 6 missing labeling such as A, B and C on the three graphs]
Response 2: Thank you for pointing this out. We agree with this comment. Therefore, we have rearranged the Figure 6 with proper labelling.
Comments 3: [Line 348 please undo the italicizing the word “and”.]]
|
|
|
|
Response 3: Thank you for pointing this out. We agree with this comment. Therefore, we have corrected the typo.
|

Reviewer 3 Report
Comments and Suggestions for Authors
Karaman and colleagues present a thorough and well-structured study on pathway-specific insights into colorectal cancer using a multi-omics data integration approach. The methodology is robust, and the findings provide valuable insights into host-microbiome interactions in colorectal cancer progression. However, several areas could be improved in terms of clarity, structure, and data interpretation.
- The abstract effectively summarizes the study but could be more concise. Consider refining the wording to enhance clarity, particularly in explaining the major findings
- The statement on page 2, "Only one omics analysis is not enough to evaluate the diseases," should be rephrased for clarity. A more precise alternative is: "A single-omics approach is insufficient to comprehensively evaluate complex diseases such as cancer."
- Figure 4: Adding annotations to clarify key clusters and relationships would enhance interpretability
- Figure 6: The description could be improved to better explain how different datasets contribute to distinguishing CMS subgroups
The manuscript is well-written and presents complex multi-omics analyses in a structured manner.
Author Response
|
Comments 1: [The abstract effectively summarizes the study but could be more concise. Consider refining the wording to enhance clarity, particularly in explaining the major findings]
Response 1: Thank you for pointing out your valuable comments on our paper. We agree with this comment. Therefore, we have enhanced the clarity of the findings in a more concise way.
Comments 2: [The statement on page 2, "Only one omics analysis is not enough to evaluate the diseases," should be rephrased for clarity. A more precise alternative is: "A single-omics approach is insufficient to comprehensively evaluate complex diseases such as cancer."]
Response 2: Thank you for pointing this out. We agree with this comment. Therefore, we have rearranged sentence as suggested. [A single-omics approach is insufficient to comprehensively evaluate complex diseases such as cancer [7,8].]
Comments 3: [Figure 4: Adding annotations to clarify key clusters and relationships would enhance interpretability]
Response 3: Thank you for pointing this out. The heatmap was constructed to visualize the correlation of all variables without pre-filtering steps. In most of the integration methods, filtering the data is quite necessary to get rid of the noisiness in the datasets. However, we believe that there might be still some correlations between filtered out variables. The aim of introducing this result is to indicate association of all variables including non-filtered ones. After obtaining non-neglectable results from the analysis, we only chose to focus on the correlation between variables. Because the noisiness of the data causes complex clustering which does not let us comment on clear discrimination of CMS subgroups. It may be considered in case of having large number of sample sizes. However, the only focus in this part was to be able to catch the correlation between non-filtered ones as we managed.
Comments 4: [Figure 6: The description could be improved to better explain how different datasets contribute to distinguishing CMS subgroups.]]
|
|
Response 4: Thank you for pointing this out. We agree with this comment. Therefore, we have improved the description of Figure 6 as you suggested. [Figure 6. The variables were selected by N-Integration model and the most explanatory contributors that constructed component 1 obtained from preprocessed A) transcriptome, B) microbiome, and C) microbial pathway datasets are indicated in figure (blue represents CMS1, orange represents CMS2, and gray represents CMS3).] The details about the explanatory power of selected variables were already given in the text right before Figure 6. Moreover, the power of using multi-dataset was pointed out in PCA results. It demonstrates that using single-omic approach does not always provide enough informative variables however obtaining other explanatory variables from different datasets provides the better discrimination of groups of samples.
|
|
4. Response to Comments on the Quality of English Language |
|
Point 1: The English could be improved to more clearly express the research. |
|
Response 1: Quality of English was improved from beginning to end as suggested by the reviewer. |

Reviewer 4 Report
Comments and Suggestions for Authors
In this paper, Karaman et al. attempt to use multi-omics data analysis to better understand interactions in complex diseases, specifically colorectal cancer in this case.
Other specific critiques that I've noticed are:
- Lines 62-64 mentioned some previous research, but a citation is missing here.
- In the Materials and Methods section, lines 108-109, it is not clear if the tumor samples mentioned were from the previously published paper from which the data sets were pulled, or elsewhere.
- For the CMS groups, please add more details about how these groups were determined and allotted.
- For Figure 1, the inclusion of the advantages of the two strategies feels odd in a flow chart.
- The legend for figure one has a typo.
- For figure 2, please make all the Y axes the same. This will enable a visual comparison of the plots to each other.
- Table 1 is missing a legend.
- The volcano plot in Figure 3 can be made more visually appealing by changing the Y axis.
- In the various sections for the results as well as discussion, please try to connect the data set and your findings to the findings of the original paper that the data belongs to.
Comments on the Quality of English Language
I did find quite a few language issues and grammatical errors throughout the manuscript. There were too many to list, but specifically issues were found in lines 16-18, 21-23, 32, 40, 49,58, 68, 78, 79, 86, 87, etc. I did stop keeping track afterwards, so, request the authors to go through the entire manuscript to make corrections in all the sections. I also observed that most sections can use some more conciseness, especially in how the ideas and the conclusions are presented.
Author Response
|
Comments 1: [Lines 62-64 mentioned some previous research, but a citation is missing here.] Response 1: Thank you for pointing out your valuable comments on our paper. We totally agree with this comment. The following sentence was also about the same study and the citation was put after the second sentence. However, it totally right to use the citation in the first sentence as you suggested. [In research conducted by Vartika Bisht and his group, RFE, the Bayesian additive regression trees method, and Bayesian classification were used for the integration of omics datasets, and they discovered that the Kiaa1199, Cdh3, Guca2b, Lgals4, Ca7, Nrc3c2, Abcg2, and Aqp8 genes were associated with cancer pathophysiology [19].]
Comments 2: [In the Materials and Methods section, lines 108-109, it is not clear if the tumor samples mentioned were from the previously published paper from which the data sets were pulled, or elsewhere.] Response 2: Thank you for your valuable information. We agree with this comment. Therefore, we have corrected the sentence as [In that study, tumor samples were collected from 34 colorectal cancer patients (20 female, 14 male); the age ranged from 44 to 88 years (mean: 74 ± 8.4).]
Comments 3: [For the CMS groups, please add more details about how these groups were determined and allotted.] Response 3: Thank you for pointing this out. We agree with this comment. Therefore, we have enriched the introduction part with following information. [Consensus Molecular Subtypes (CMS) of colorectal cancer (CRC) are stratified based on key biological characteristics, including immune infiltration and activation, signaling pathway dynamics, metabolic dysregulation, and stromal involvement. These subtypes provide critical insights into the heterogeneity of CRC, reflecting differences in tumor biology, patient prognosis, and therapeutic response. While transcriptomic profiling remains the predominant method for CMS classification, emerging research is exploring microbiome-based predictive models, leveraging host-microbiome interactions to infer CMS groups with potential implications for precision oncology.]
Comments 4: [For Figure 1, the inclusion of the advantages of the two strategies feels odd in a flow chart.] Response 4: Thank you for pointing this out. In general, the variables are filtered out for the multi-omics data integration analysis to get rid of the noisiness of the datasets. Because analysis of huge datasets including hundreds of variables is not easy to handle, it causes complexity in many cases. However, we wanted to point out that the integration of non-filtered data can still provide valuable outputs. So, we performed appropriate integration methods with non-filtered and filtered datasets and discussed the results. At the end, we thought it would be good to demonstrate it with a scheme including the flow and advantages of the approaches we used.
Comments 5: [The legend for figure one has a typo.] Response 5: Thank you for pointing this out. We agree with this comment. Therefore, we have corrected the typo. [Figure 1. The chart demonstrates the experimental flow and advantages of both approaches at the end].
Comments 6: [For figure 2, please make all the Y axes the same. This will enable a visual comparison of the plots to each other.] Response 6: Thank you for pointing this out. We agree with this comment. Therefore, we have revisualized the boxplot as suggested. New Figure 2 provides the comparison of the mean relative abundance to each other.
Comments 7: [Table 1 is missing a legend.] Response 7: Thank you for pointing this out. We agree with this comment. Therefore, we have corrected the legend. [Table 1. Pairwise Wilcoxon results are indicated in table.]
Comments 8: [The volcano plot in Figure 3 can be made more visually appealing by changing the Y axis.] Response 8: Thank you for pointing this out. There are two significant points of analysing our transcriptome data. The first one is to obtain host-pathway information, and the second one is to demonstrate PCA construction to prove that only one single-omic is not enough to explain the experimental groups (CMS groups in our case). We chose to focus on these two parts as we pointed out and discussed in our paper. That is the reason why we did not dig deep into analysis of microbiome and transcriptome data, so we attract attention mostly to the integration part. Moreover, these analyses were already studied well in previous papers, we thought that repeating the same information is pointless.
Comments 9: [In the various sections for the results as well as discussion, please try to connect the data set and your findings to the findings of the original paper that the data belongs to.] Response 9: Thank you for pointing this out. The aim of our study and the aim of the previous study was quite different from each other. This is the reason why we chose not to focus on comparing our results. In that study, microbiome data was used to determine CMS subgroups which are classified based on RNA-Seq data. They used transcriptomic data to check how accurately the samples were grouped. In our study, we used these omics datasets to get pathway information and integrated variables obtained from each dataset to discover a potential association to make prediction about colorectal cancer disease mechanism of action.
|
|
4. Response to Comments on the Quality of English Language |
|
Point 1: I did find quite a few language issues and grammatical errors throughout the manuscript. Request the authors to go through the entire manuscript to make corrections in all the sections. |
|
Response 1: Quality of English was improved from beginning to end as suggested by the reviewer. |

Round 2
Reviewer 4 Report
Comments and Suggestions for Authors
Technical Issues:
- Please review the manuscript again as there are still plenty grammar issues. I also notice an overuse of ‘the’ and commas in this version.
- In specific issues, line 23, the word ‘chaotic’ here isn’t giving it the intended meaning
- Same with line 24, ‘molecular level of action’
- Same with line 28, ‘was also found to play’
- Line 49, should be ‘complex diseases like cancer’, instead of ‘cancer-like complex disease’
- Line 63, I believe Vartika Bhisht is a female, and instead of saying ‘Vartika Bisht and his group’, it might be best to say ‘Bisht et al’
- Line 65, Unnecessary usage of ‘the’
- Line 70, I believe here the use of the term cancer is incorrect as the data and findings were for colorectal cancer specifically.
- Line 102, ‘ncbi’ should be ‘NCBI’
- Line 108, should be non-negligible
- Line 114, should be ‘titled’ instead of ‘entitled’
- Lines 118 and onwards, ‘site’, ‘side’, etc should be in single quotes
- Line 165, the package is called ‘pathfindR’
- For results section 3.1, please write about when a finding is in a figure or not shown
- Line 275, Unnecessary usage of ‘the’
- In the discussion section add some more background/citations as it presently looks like an expanded results section.
Basic grammar issues and hard to understand terminology that doesn't support the intended meaning throughout the manuscript
Author Response
|
Comments 1: [Please review the manuscript again as there are still plenty grammar issues. I also notice an overuse of ‘the’ and commas in this version.] Response 1: Thank you for pointing out your valuable comments on our paper. We agree with this comment. Manuscript were reviewed by following your suggestions
Comments 2: [In specific issues, line 23, the word ‘chaotic’ here isn’t giving it the intended meaning] Response 2: Thank you for your valuable information. We agree with this comment. Therefore, we have corrected the sentence as [Thousands of biomarkers have been discovered to solve the mechanisms of cancer, but dynamic alterations in the parameters that affect cancer progression cause complex disease status.]
Comments 3: [Same with line 24, ‘molecular level of action’] Response 3: Thank you for pointing this out. We agree with this comment. Therefore, we have changed the word “chaotic” into “complex”. [Therefore, it is essential when dealing with cancer to analyze all parameters, including pathway information, to understand the disease mechanism of action.]
Comments 4: [Same with line 28, ‘was also found to play’] Response 4: Thank you for pointing this out. We agree with this comment. Therefore, we have written the sentence in a more advanced and understandable way. [The Klf3 gene has been identified as a critical regulator in the activation of the WNT1 and WNT/β-catenin signaling pathways. Notably, it exhibited a strong positive correlation with the presence of Fusobacteria, which are also implicated in modulating these pathways.]
Comments 5: [Line 49, should be ‘complex diseases like cancer’, instead of ‘cancer-like complex disease’] Response 5: Thank you for pointing this out. We agree with this comment. Therefore, we have corrected the sentence as suggested. [Because, complex diseases like cancer need the combination of the omics study outputs to provide an evaluation of a disease from all perspectives [9].]
Comments 6: [Line 63, I believe Vartika Bhisht is a female, and instead of saying ‘Vartika Bisht and his group’, it might be best to say ‘Bisht et al’] Response 6: Thank you for pointing this out. We agree with this comment. Therefore, we have corrected the sentence as suggested. [In research conducted by Bisht et. al., RFE, the Bayesian additive regression trees method, and Bayesian classification were used for the integration of omics datasets, and they discovered that Kiaa1199, Cdh3, Guca2b, Lgals4, Ca7, Nrc3c2, Abcg2, and Aqp8 genes were associated with cancer pathophysiology [19]
Comments 7: [Line 65, Unnecessary usage of ‘the’] Response 7: Thank you for pointing this out. We agree with this comment. Therefore, we have deleted unnecessary “the” from the sentence as suggested. [In research conducted by Bisht et. al., RFE, the Bayesian additive regression trees method, and Bayesian classification were used for the integration of omics datasets, and they discovered that Kiaa1199, Cdh3, Guca2b, Lgals4, Ca7, Nrc3c2, Abcg2, and Aqp8 genes were associated with cancer pathophysiology [19]
Comments 8: [Line 70, I believe here the use of the term cancer is incorrect as the data and findings were for colorectal cancer specifically.] Response 8: Thank you for pointing this out. We totally agree with this comment. Therefore, we have specified it as “colorectal cancer” in the sentence. [All findings have shown remarkable colorectal cancer hallmarks, such as resistance to apoptosis, angiogenesis, and cell evasion, increasing our understanding of molecular level effects on the progression of colorectal cancer.]
Comments 9: [Line 102, ‘ncbi’ should be ‘NCBI’] Response 9: Thank you for pointing this out. We totally agree with this comment. Therefore, we have corrected it as “NCBI”. [The results of the integration were consolidated by the host pathway dataset obtained from transcriptome data and databases such as bencard, MetaCyc, NCBI, KEGG (Kyoto Encyclopedia of Genes and Genomes) and UniProt (Universal Protein Knowledgebase) [25-29].]
Comments 10: [Line 108, should be non-negligible] Response 10: Thank you for pointing this out. We totally agree with this comment. Therefore, we have corrected it as suggested. [Both approaches presented non-negligible outcomes by providing latent interactions between the datasets.]
Comments 11: [Line 114, should be ‘titled’ instead of ‘entitled’] Response 11: Thank you for pointing this out. We totally agree with this comment. Therefore, we have corrected it as suggested. [Transcriptome and microbiome 16S rRNA raw datasets were obtained from a re-cent study titled “Distinct gut microbiome patterns associate with consensus molecu-lar subtypes of colorectal cancer” published by Rachel V. Purcell et. al. in 2017 [24].]
|
|
|
|
Comments 12: [Lines 118 and onwards, ‘site’, ‘side’, etc should be in single quotes] Response 12: Thank you for pointing this out. We totally agree with this comment. Therefore, we have corrected it as suggested. [Beside age and gender, the metadata also contained ‘site’ data, indicating where the tumor sample had been taken from (1 rectal sample and 33 colonic samples), ‘side’ da-ta, referring to the side of the body the tumor sample was taken from (21 from right-side and 12 from left-side), ‘stage’ data, referring to the stage of the cancer pa-tient (5 stage I, 14 stage II, 13 stage II, and 1 stage IV), and ‘CMS’ data (6 in the CMS1 group, 13 in the CMS2 group, 9 in the CMS3 group, and 5 unclassified (UC) samples).] |
|
|
|
Comments 13: [Line 165, the package is called ‘pathfindR’] Response 13: Thank you for pointing this out. We totally agree with this comment. Therefore, we have corrected it as suggested. [Raw read counts were normalized to transcripts per million (TPM) using the DESeq2 package (v.1.10.1) in R. RNA-seq analysis culminated in pathway enrichment analysis using the ‘pathfindR’ R package (v.2.1.0) in R [33].]
Comments 14: [For results section 3.1, please write about when a finding is in a figure or not shown] Response 14: Thank you for pointing this out. It is specified as you suggested. [The results of the 16S rRNA microbiome analysis indicated that CRC patients that belonged to the CMS1 group had enriched levels of Fusobacteria (p < 0.05) and Bacteroidetes (p > 0.05) and decreased levels of Firmicutes (p > 0.05) and Proteobacteria (p < 0.05) at the phylum level (p < 0.05) (Figure 2). Although genus-level results were not visualized, our analysis revealed a significant increase in the relative abundance of Prevotella in the CMS1 group, whereas Bacteroides was markedly enriched in the CMS2 and CMS3 groups.]
|
|
|
|
|
|
Comments 15: [Line 275, Unnecessary usage of ‘the’] Response 15: Thank you for pointing this out. We totally agree with this comment. Therefore, we have corrected it as suggested. [A volcano plot was constructed using the results generated from RNA-seq analysis. The Rn7sk, Rpph1, Rn7sl1, Rn7sl2, Znf460, Snord17, Snora73b, H1-4, H4c12, and H3c7 genes were discovered to be highly upregulated in the tumor tissue collected from colorectal cancer patients (Figure 3A).]
|
|
|
|
|
|
Comments 16: [In the discussion section add some more background/citations as it presently looks like an expanded results section.] Response 16: Thank you for pointing this out. We have enriched the discussion part by including 7 more citations as you suggested.
|
